# PROVABLY COMMUNICATION-EFFICIENT FEDERATED GRAPH NEURAL NETWORK

## ABSTRACT

Graph neural networks (GNNs) are powerful tools for relational data, but their application is often limited by data silos and privacy concerns, as real-world graphs are frequently distributed across multiple clients. While federated learning (FL) offers a privacy-preserving training paradigm, existing federated GNN approaches suffer from a critical flaw: they either ignore the crucial links between clients, sacrificing accuracy, or require impractically high communication overhead. We introduce CE-FedGNN, a communication-efficient federated GNN framework for such coupled graphs. Instead of sharing raw data or per-iteration embeddings, CE-FedGNN infrequently transmits only aggregated, high-level embeddings, preserving critical structural context while minimizing privacy leakage and communication costs. Despite the challenges of optimization under multi-layer composition and coupled data, we establish a convergence rate of $O(1/\sqrt{T})$ to a stationary point while the communication complexity is $O(T^{3/4})$. We further derive bounds for injecting Gaussian noise that provide formal differential privacy. Our experiments on a synthetic interbank anti-money laundering task show that the effectiveness of CE-FedGNN, which can be preserved even with injected Gaussian noise for differential privacy.

## 1 INTRODUCTION

Graph neural networks (GNNs) have become a leading paradigm for learning from relational data, achieving state-of-the-art results in areas such as social network analysis, traffic forecasting, and financial fraud detection (Shu et al., 2019; Derrow-Pinion et al., 2021; Egressy et al., 2024). By propagating and aggregating information through graph structures, GNNs capture complex dependencies and emergent patterns that are invisible to traditional methods. However, most existing approaches assume centralized access to the entire graph, which is an assumption that is rarely valid in practice due to strict privacy regulations (e.g., GDPR, CCPA) and competitive business barriers. For example, in cross-bank money laundering detection, financial institutions each observe only a fragment of the global transaction graph and cannot share raw data with others.

Federated learning (FL) provides a natural framework for collaborative model training without centralizing sensitive data. Standard FL methods such as FedAvg (Konečný et al., 2016; McMahan et al., 2017) are designed for independent local datasets, where the global objective is simply the average loss across clients. Graph data violates this assumption: edges connect nodes across clients, and a node's neighborhood often spans multiple institutions. This cross-client coupling makes neighborhood aggregation, which is a core GNN operation, dependent on external information. Ignoring cross-client edges leads to incomplete and inaccurate models, while simply sharing the necessary embeddings at every iteration introduces prohibitive communication costs. Moreover, the multi-layer compositional structure of GNNs amplifies noise and bias from partial information, further complicating optimization. These challenges are especially acute in applications such as financial crime detection, where malicious behaviors such as laundering cycles or "smurfing" schemes are emergent properties of the global graph. No single bank can detect such patterns in isolation, underscoring the need for communication-efficient, privacy-preserving methods that faithfully capture cross-institutional dependencies. An illustration has been presented in Figure 1(a).

Existing federated GNN methods present a stark trade-off between modeling accuracy and communication efficiency. One line of work simplifies the problem by entirely ignoring cross-client edges

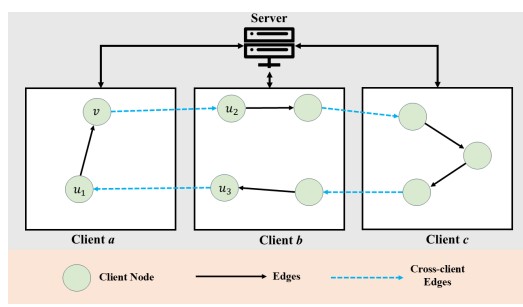 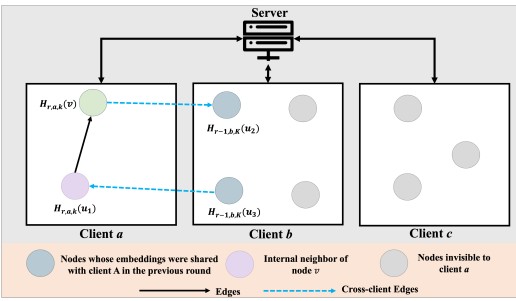

(a) Illustration of the challenge in FedGNN: a cycle that cannot be seen by either client alone.

(b) When the edge $(v, u_2)$ is sampled on client a, it uses recent shared embedding node $u_2$ from client b.

Figure 1: Illustration of the challenge in FedGNN and our algorithmic design.

(Du et al., 2022; Peng et al., 2022; Scardapane et al., 2020; He et al., 2021; Pan et al., 2023; Hu et al., 2022), sacrificing the model's ability to capture critical global graph properties. Conversely, methods that do account for these edges (Chen et al., 2021; Wu et al., 2022; 2021) typically require the costly exchange of node embeddings in every training iteration. A middle-ground approach shares embeddings only once at initialization (Yao et al., 2023), but this fails to adapt to representation drift during training, leading to suboptimal accuracy. Sharing embeddings only once at initialization fails to track representation drift and degrades accuracy. (Du & Wu, 2022) reduces communication by infrequently sharing node embeddings. However, this approach overlooks nodes not sampled in the most recent global round and requires feature sharing, which raises privacy concerns. Moreover, its theoretical guarantees only ensure convergence to a neighborhood of the solution. Another method (Qiu et al.) uses global connections infrequently but relies primarily on local graphs, which may still obscure important global patterns. (Guo et al., 2023) addresses cross-client coupling in federated compositional optimization via decomposing the gradient into active/passive parts, where active parts depend on local data and passive parts depend on other clients. The summary statistics are shared every $K$ steps. While provably communication-efficient, their framework cannot apply to GNNs, which have a multilayer compositional structure and interdependence between inner data and outer. What is more, the above methods lack a formal privacy guarantee. In federated GNNs, sharing aggregated embeddings is less sensitive than sharing individual embeddings Yao et al. (2023), however, recent work has shown that sharing intermediate embeddings in GNNs can still leak sensitive information (Zhang et al., 2024; Duddu et al., 2020; Zhang et al., 2022; Li et al., 2020).

To this end, we design a communication-efficient FedGNN algorithm and analyze the communication and iteration complexity with and without differential privacy. We study a realistic setting where an edge between two clients is visible to both parties (i.e., edge attributes are shared), while node attributes remain private. For example, when a transaction occurs between two banks, both banks observe its attributes (e.g., time, amount), but each retains only its own node features. Our contributions are as follows.

- **Decomposition framework for federated GNNs.** We design a tailored decomposition scheme that maintains a moving-average estimator of node embeddings to mitigate the variance and bias from mini-batch updates. Crucially, this mechanism is applied only to nodes (not edges), as edges can vastly outnumber nodes in practice. When a neighbor lies on another client, we use its most recently shared moving-average embedding. We prove this provides sufficiently accurate estimates by explicitly accounting for the resulting latency error. This allows us to restrict cross-client interactions to 1-hop neighbors and avoid multi-hop sampling. The framework, illustrated in Figure 1(b), further reduces privacy risks by transmitting only high-level embeddings instead of intermediate representations from every layer. We also extend the moving-average strategy to gradients. With $T$ iterations, our algorithm achieves a convergence rate of $O(1/\sqrt{T})$ toward a stationary point while requiring only $O(T^{3/4})$ communication rounds.

- **Differential privacy mechanisms.** In addition to conventional noise injection into model parameters and gradients, we introduce Gaussian noise into shared embeddings. We analyze the convergence rate under the injected noise.

- **Empirical evaluation.** Experiments on a synthetic anti–money laundering task demonstrate the effectiveness and communication efficiency of our method. Furthermore, with Gaussian noise injection, we provide differential privacy, and our experiments confirm that strong utility is retained even under significant noise levels.

## 2 RELATED WORK

**Graph Neural Networks.** Unlike images or text, many real-world problems involve entities connected by complex relationships that cannot be naturally represented in Euclidean space. Graph neural networks (GNNs) provide a principled way to handle such data by propagating and aggregating information across graph structures. They have shown remarkable success in domains such as social networks (Badrinath et al., 2025; Hou et al., 2025), transportation systems (Derrow-Pinion et al., 2021; Zheng et al., 2023), simulation (Sanchez-Gonzalez et al., 2020; Jain et al., 2025), and biological systems (Bongini et al., 2021). Beyond these, GNNs are particularly advantageous for reasoning tasks that require capturing structural dependencies, including combinatorial optimization (Dudzik & Veličković, 2022; Bevilacqua et al., 2023) and financial fraud detection (Egressy et al., 2024; Lin et al., 2024), highlighting scenarios where expressive GNN architectures are particularly crucial. More discussion of recent advances of GNN can be found in Appendix A.

**Communication-Efficient Federated Learning.** A primary challenge in Federated Learning (FL) is communication overhead. A vast body of work addresses this for problems where the global objective is a separable average of individual client losses (Konečný et al., 2016; McMahan et al., 2017; Stich, 2018; Yu et al., 2019a;b; Yang, 2013; Karimireddy et al., 2020; Kairouz et al., 2021; Khaled et al., 2020; Woodworth et al., 2020b;a; Haddadpour et al., 2019; Deng & Mahdavi, 2021; Deng et al., 2020; Liu et al., 2020; Sharma et al., 2022; Li et al., 2022; Huang et al., 2022; Tarzanagh et al., 2022; Xing et al., 2022).

However, these conventional FL algorithms struggle with coupled objectives where client data is interdependent. Some works address specific forms of coupling: Yuan et al. (2021); Guo et al. (2020) study AUC maximization by reformulating it into a decomposable minimax problem, though this approach lacks generality. Gao et al. (2022) provides convergence analysis for compositional problems but ignores cross-client coupling. Guo et al. (2023) propose a communication-efficient method for a general pairwise objective using an active-passive gradient decomposition. While a step forward, their two-level compositional setup assumes inner and outer data are independent, making it inapplicable to the complex dependencies in federated GNNs. Overall, many existing solutions are largely ad hoc and lack strong theoretical foundations (Han et al., 2022; Zhang et al., 2020; Wu et al., 2022; Li & Huang, 2022).

Particularly for federated GNNs, which are subject to the challenge of cross-client edges, most existing approaches either ignore these edges, (Du et al., 2022; Peng et al., 2022; Scardapane et al., 2020; He et al., 2021). or incur prohibitive communication costs by requiring clients to share node embeddings every training iteration (Wu et al., 2022; 2021). Some works attempt to reduce this cost by heuristics utilizing global information sparsely Yao et al. (2023); Qiu et al..

**Privacy in Federated Learning.** While the above methods on federated compositional problems aim to reduce communication overhead, they largely overlook the privacy implications of sharing high-dimensional embeddings. Differential privacy (DP) has become the dominant formal approach for limiting information leakage in FL (Abadi et al., 2016; McMahan et al., 2018; Truex et al., 2020; Wei et al., 2020). In federated GNNs, very few formally analyze privacy guarantee, while intermediate representations can encode sensitive information about nodes and their neighborhoods, especially in financial or social domains.

## 3 METHOD

We now formalize the problem, introduce our communication-efficient federated GNN algorithm, and extend it with Gaussian perturbations to ensure differential privacy.

### 3.1 PROBLEM STATEMENT

A large class of modern GNNs is formulated under the Message Passing framework (Wu et al., 2020). The central mechanism is to iteratively aggregate information from neighbors and update node representations. Consider a graph $\mathcal{G} = (\mathcal{V}, \mathbb{E})$ with node set $\mathcal{V}$ and edge set $\mathbb{E}$. In each layer $l$, a node $v$ aggregates messages from its neighbors $\mathcal{N}(v)$ and updates its representation:

$$\mathbf{h}^{(l)}(v) = \text{UPDATE}^{(l)}\left(\mathbf{h}^{(l-1)}(v), \text{AGGREGATE}^{(l)}(\{\mathbf{h}^{(l-1)}(u) : u \in \mathcal{N}(v)\})\right) \tag{1}$$

where the AGGREGATE function is typically permutation-invariant (e.g., summation or mean).

In this work, we focus on a common setup in which the aggregation function is chosen from GCN(Kipf & Welling, 2017), GraphSAGE-mean(Hamilton et al., 2017), or GIN(Xu et al., 2018), and the update function sums the node's previous embedding with the aggregated message, followed by a learnable transformation (e.g., a linear layer and nonlinearity). Without loss of generality, we adopt GraphSAGE-mean, which can be written as:

$$\mathbf{h}^{(l)}(v) = \phi\left(\mathbf{W}^{(l)} \cdot \frac{1}{|\mathcal{N}(v) \cup v|} \sum_{u \in \mathcal{N}(v) \cup v} \mathbf{h}^{(l-1)}(u)\right), \tag{2}$$

where $\phi$ denotes a nonlinear activation function. We let $\hat{h}^l(v) := \sum_{u \in \mathcal{N}(v) \cup v} \mathbf{h}^{(l-1)}(u)$, $\tilde{h}^l(v) := \mathbf{W}^{(l)}\hat{h}^l(v)$ and thus $\mathbf{h}^{(l)}(v) = \phi(\tilde{h}^l(v))$. We simplify the notation $\mathcal{N}(v) \cup v$ to $\mathcal{N}(v)$ when the context is clear. The formulations of GCN and GIN are provided in Appendix A.

For node- or edge-level prediction, a linear layer is applied to the final embeddings:

$$\hat{y}_x = F(\mathbf{W}^{L+1}; \mathbf{h}(x)), \quad \mathcal{L}(x) = \ell(\hat{y}_x, y_x), \quad x \in \mathcal{X}, \tag{3}$$

where $\mathcal{X} = \mathcal{V}$ for the node classification task and $\mathcal{X} = \mathcal{E}$ for the edge classification task, and edge representations are further computed via mean aggregation of their endpoints. For an edge $e = (u, v)$, we have

$$\mathbf{h}(e) = \phi\left(\mathbf{W}^e \cdot \frac{\mathbf{h}^{(L)}(u) + \mathbf{h}^{(L)}(v)}{2}\right). \tag{4}$$

In federated learning, with $\mathbf{W}$ denoting the model of all layers, the formulation is

$$F(\mathbf{W}) = \frac{1}{N} \sum_{i=1}^{N} F_i(\mathbf{W}), \quad \text{where} \quad F_i(\mathbf{W}) = \frac{1}{|\mathcal{X}_i|} \sum_{x \in \mathcal{X}_i} \mathcal{L}(x). \tag{5}$$

### 3.2 A COMMUNICATION-EFFICIENT ALGORITHM

To illustrate the challenge, consider computing gradients on a given client. For instance, the gradient of $F_i(\mathbf{W})$ with respect to $\mathbf{W}^e$ for an edge $e = (u, v)$, where node $v$ lies outside client $i$, is

$$\frac{\partial F_i(\mathbf{W}; e)}{\partial \mathbf{W}^e} = \left(\frac{\partial F_i(\mathbf{W}; e)}{\partial h(e)} \phi'\left(\mathbf{W}^e \cdot \frac{\mathbf{h}^{(L)}(u) + \mathbf{h}^{(L)}(v)}{2}\right) \frac{\mathbf{h}^{(L)}(u) + \mathbf{h}^{(L)}(v)}{2}\right). \tag{6}$$

The highlighted term depends on the embedding of a cross-client neighbor. This leads to several challenges: 1) **Cross-client dependency.** Local clients cannot compute the neighbor embedding $\mathbf{h}^{(L)}(v)$ on their own; 2) **High communication cost.** Naively exchanging embeddings at every iteration incurs prohibitive communication overhead; 3) **Gradient backpropagation.** By the chain rule, gradients must also flow through the cross-client embedding $\mathbf{h}^{(L)}(v)$, which further complicates local updates. 4) **Biased gradients.** Even with embedding exchange, estimating $h^{(l-1)}(u)$ from minibatches produces bias due to the nonlinear composition of $\phi(\cdot)$ and $\phi'(\cdot)$.

We propose to maintain *moving averages* of both (i) intermediate embeddings and (ii) gradient estimators, which provide low-variance approximations. Let subscript $r, i, k$ denote round $r$ on client $i$ at iteration $k$. For clarity, we first assume that a node's neighbors are restricted to those residing in the same client, while cross-client interactions occur only through inter-client edges. (In Appendix F, we extend the discussion to cases where neighbors of nodes may span multiple clients.)

The forward pass for a node $v$ with a moving-average update is defined as

$$\tilde{H}_{r,i,k}^{(l)}(v) = (1 - \gamma)\tilde{H}_{r,i,k-1}^{(l)}(v) + \gamma \frac{1}{n_{r,i,k}(v)} \mathbf{W}_{r,i,k-1}^{(l)} \cdot \sum_{u \in \mathcal{N}_{r,i,k}(v)} H_{r,i,k}^{(l-1)}(u),$$

$$H_{r,i,k}^{(l)}(v) = \phi(\tilde{H}_{r,i,k}^{(l)}(v)),$$

(7)

where $n_{r,i,k}(v) = |\mathcal{N}_{r,i,k}(v)|$, $\mathcal{N}_{r,i,k}(v)$ are local neighbors in the batch, and the input embeddings satisfy $H^{(0)}(\cdot) = h^{(0)}(\cdot)$. We further denote $\mathbf{W}_{r,k} := \frac{1}{N} \sum_i \mathbf{W}_{r,i,k}$.

The behavior of this estimator is captured by the following lemma:

**Lemma 3.1.** *Under appropriate conditions, with $p$ being the smallest probability for a node to be sampled, the Algorithm 1 ensures that*

$$\frac{1}{RK} \sum_R \sum_k \|H_{r,i,k}^{(l)}(u) - h^{(l)(u)}\|^2 \leq O(\frac{1}{\gamma p R K} + \frac{\gamma}{p} + \gamma^2 K^2 + \frac{\|\mathbf{W}_{r,k} - \mathbf{W}_{r,k-1}\|^2}{\gamma^2}).$$

(8)

**Remark.** By appropriately choosing parameters $\gamma$ and $K$, the first three error terms decrease with the number of rounds $R$. To control the final term, we next introduce a gradient estimator that ensures model updates are computed from low-variance estimates. Consequently, the variance of the embedding estimator $H$ converges along with the gradient estimator and true gradient. Importantly, only 1-hop node embeddings need to be communicated across clients, since each node maintains a moving average of its embeddings. This design eliminates the need for multi-hop cross-client sampling.

The batch stochastic gradient over $W_e$ is

$$\hat{\nabla} F_i(\mathbf{W}_{r,i,k}^e; B_{r,i,k}) = \left( \frac{\partial F_i(\mathbf{W}_{r,i,k}^e; e; B_{r,i,k})}{\partial h(e)} \phi'\left(\mathbf{W}_{r,i,k}^e \cdot \hat{h}_{r,i,k}(e)\right) \hat{h}_{r,i,k}(e) \right).$$

(9)

where

$$\hat{h}_{r,i,k}(e) = \frac{\mathbf{h}_{r,i,k}^{(L)}(u) + \mathbf{h}_{r-1,c(v),K}^{(L)}(v)}{2},$$

(10)

with $c(v)$ denoting the host client of node $v$. Here, $B_{r,i,k}$ denotes a minibatch sampled on client $i$, and the subscript $H$ of $F$ indicates that moving-average estimators are applied in the forward pass. The key difficulty is that the gradient depends on late-layer embeddings from other clients (highlighted in blue). By the chain rule, backpropagation requires differentiating through this term, which a local client cannot perform.

We resolve this issue by observing that the edge $e = (u, v)$ also exists in client $c(v)$. Without loss of generality, we assume $e$ is sampled with equal probability on both $c(u)$ and $c(v)$; otherwise, one can reweight the sampling or explicitly instruct $c(v)$ to sample $e$ in the next round. The gradient decomposition on client $c(v)$ is symmetric, therefore, client $c(v)$ can compute gradients through the blue term, while client $c(u)$ only needs to provide embeddings of $u$. As a result, although each client computes only part of the gradient, their aggregated contributions, $\frac{1}{N} \sum_i \hat{\nabla} F_i(\mathbf{W}_{r,i,k}^e; B_{r,i,k})$, recover the full global gradient.

Then we apply a moving average for the gradient estimation:

$$G_{r,i,k}^{(l)} = (1 - \beta)G_{r,i,k}^{(l)} + \beta \hat{\nabla} F_i(\mathbf{W}; B_{r,i,k}),$$

(11)

whose behavior is analyzed in the following lemma:

**Lemma 3.2.** *Under appropriate conditions, with $\bar{G}_{r,k} = \frac{1}{N} \sum_i G_{r,i,k}$ the Algorithm 1 ensures that*

$$\mathbb{E}\|\bar{G}_{r,k} - \nabla F(\mathbf{W}_{r,k})\|^2 \leq O\left( \frac{1}{\beta R K} + \frac{1}{\gamma p R K} + \frac{\beta}{N} + \frac{\gamma}{p} + \beta^2 K^2 + \frac{\eta^2}{\beta^2}\|\nabla F(\mathbf{W}_{r,k-1})\|^2 \right).$$

---

**Algorithm 1** CE-FedGNN

---

**On Server:**
Initialization: $\mathbf{W}^{(l)}, \forall l$, global buffer of embeddings $\mathcal{H}$
**for** $r \in [1, 2, .., R]$ **do**
    Collect requests of neighbor embeddings from all clients, and route them to corresponding
    clients, collect those embeddings from those banks, and send them to clients.
    Clients in parallel do:
        $\mathbf{W}_{r,i}, \mathcal{H}_{r,i} \leftarrow \text{LocalUpdate}(\mathbf{w}_{r,i})$
    $\mathbf{W}_r = \frac{1}{N} \sum\limits_{i \in [N]} \mathbf{W}_{r,i,K}, G_r^{(l))} = \frac{1}{N} \sum\limits_{i \in [N]} G_{r,i,K}^{(l)}$
    $\mathbf{W}_{r,i,0} = \mathbf{W}_r, G_{r,i,0} = G_r$
    Flush $\mathcal{H}$ with $\mathcal{H}_{r,i}$
**end for**
Output: $\mathbf{w}^{(E+1,0)}$.

---

**On Local Client W**:
Initialization: Receive $\mathbf{W}_{r,i,0}^l$ from the server. Received updated node embeddings from neighbors.
**for** $k \in [K]$ **do**
    Locally sample a mini-batch $\mathcal{B}$ as 0-hop seeds, then sample $L$-hop.
    Update edge and node embeddings using (40)
    Update gradient estimator using (11)
    Update model:
$$\mathbf{W}_{r,i,k} = \mathbf{W}_{r,i,k-1} - \eta G_{r,i,k}. \tag{12}$$
**end for**
Return $\mathbf{w}_{r,i,k}$, and updated margin node embeddings $\mathcal{H}_{r,i}$ to the server.

---

**Remark.** With proper parameter setting, the estimator $G$ becomes accurate as the model converges.

Finally, the theorem of overall convergence analysis is

**Theorem 3.3.** *Under appropriate conditions, the Algorithm 1 ensures that*

$$\frac{1}{R} \sum_r \|\nabla F(\mathbf{W}_{r-1})\|^2 \leq O(\frac{1}{\eta RK} + \frac{1}{\beta RK} + \frac{1}{\gamma pRK} + \frac{\beta}{N} + \frac{\gamma}{p} + \beta^2 K^2). \tag{13}$$

**Remark.** By setting $\gamma = O(\frac{1}{R^{2/3}})$, $\beta = O(\frac{1}{R^{2/3}})$, $\eta = O(\frac{1}{R^{2/3}})$ and $K = O(R^{1/3})$, we have $\frac{1}{R} \sum_r \|\nabla F(\mathbf{W}_{r-1})\|^2 \leq \frac{1}{R^{2/3}}$, i.e. $\frac{1}{R} \sum_r \|\nabla F(\mathbf{W}_{r-1})\|^2 \leq \frac{1}{T^{1/2}}$, where $T = RK$ and $R = T^{3/4}$. To get $\mathbb{E}[\frac{1}{R} \sum_{r=1}^R \|\nabla F(\bar{\mathbf{w}}^r)\|^2] \leq \alpha^2$, we just set $R = O(\frac{1}{\alpha^3})$, $\eta = O(\alpha^2)$, $\gamma = O(\alpha^2)$, $\beta = O(\alpha^2)$ and $K = \frac{1}{\alpha}$ to yield iteration complexity of $O(1/\alpha^4)$ and communication complexity of $O(1/\alpha^3)$. We also observe that more clients can speed up the convergence by reducing the term $\frac{\beta}{N}$ and also by increasing $p$ compared to a centralized setting. In the best case, it could provide a linear speed-up, in which case by setting $R = O(\frac{1}{\alpha^3})$, $\eta = O(N\alpha^2)$, $\gamma = O(N\alpha^2)$, $\beta = O(N\alpha^2)$ and $K = \frac{1}{N\alpha}$, the total iteration complexity becomes $O(1/N\alpha^4)$.

At the end of each round $r$, clients communicate their model parameters $\mathbf{W}_{r,i,K}$, gradient estimator $G_{r,i,K}$ and update embeddings $H$ of boundary node (i.e., those needed by other clients) to neighboring clients connected via edges. The overall procedure is summarized in Algorithm 1. For intermediate embeddings $\tilde{H}$, each client transmits only the updated embeddings of its local nodes, yielding communication cost $O(Kd)$, where $K$ is the number of local steps and $d$ is the embedding dimension. It is noticeable that the number of shared nodes is typically far smaller than the number of edges. If not updated, stale embeddings from previous rounds are reused, but moving averages ensure they are accurate in expectation. Unlike (Guo et al., 2023), our method requires no auxiliary data for shared statistics; this efficiency stems from the GNN sampling strategy and the fact that each node maintains its own moving average of embeddings.

### 3.3 DIFFERENTIAL PRIVACY EXTENSION AND ANALYSIS

We now extend our algorithm to incorporate differential privacy (DP) and analyze its convergence properties. While DP for shared models/gradients has been well studied in standard federated learning, existing analyses do not account for the compositional dependencies unique to GNNs. In our setting, the local updates of $H$ and $G$ remain unchanged, but Gaussian noise is added to $H$, $G$, and $\mathbf{w}$ whenever they are communicated across clients.

Specifically, when an external neighbor is required, the update of $H$ on a client is perturbed by Gaussian noise, and the forward pass of an edge is modified if one of its endpoints belongs to another client:

$$\hat{\nabla} F_i(\mathbf{W}_{r,i,k}^e; B_{r,i,k}) = \left( \frac{\partial F_i(\mathbf{W}_{r,i,k}^e; e; B_{r,i,k})}{\partial h(e)} \phi'\left( \mathbf{W}_{r,i,k}^e \cdot \hat{h}_{r,i,k}(e) \right) \hat{h}_{r,i,k}(e) \right). \tag{14}$$

where

$$\hat{h}_{r,i,k}(e) = \frac{\mathbf{h}_{r,i,k}^{(L)}(u) + \mathbf{h}_{r-1,c(v),K}^{(L)}(v) + \mathcal{G}(0, \sigma_0^2 I)}{2} \tag{15}$$

Local updates of $\mathbf{W}$ and $G$ remain the same as in the previous subsection, but during communication rounds, Gaussian noise is injected into their aggregated forms:

$$\mathbf{W}_{r+1,i,0} = \frac{1}{N} \sum_{i=1}^{N} \mathbf{W}_{r,i,K} + \mathcal{G}(0, \sigma_1^2 I), \tag{16}$$

$$G_{r+1,i,0} = \frac{1}{N} \sum_{i=1}^{N} G_{r,i,K} + \mathcal{G}(0, \sigma_2^2 I). \tag{17}$$

Their behavior can be bounded by the following lemmas, respectively.

**Lemma 3.4.** *Under appropriate conditions, DP version of Algorithm 1 ensures that*

$$\|H_{r,i,k}^{(l)}(e) - h^{(l)(e)}\|^2 \leq O\left( \frac{1}{p\gamma RK} + \gamma + \beta^2 K^2 + \frac{\|\mathbf{W}_{r,k}^{(l)} - \mathbf{W}_{r,k-1}^{(l)}\|^2}{\gamma} + \sigma_0^2 \right). \tag{18}$$

**Lemma 3.5.** *Under appropriate conditions, DP version of Algorithm 1 ensures that*

$$\|\frac{1}{N} \sum_{i} G_{r,i,k} - \nabla F(\mathbf{W})\|^2 \leq O\left( \frac{1}{\gamma RK} + \gamma + \beta^2 K^2 + \frac{\|\mathbf{W}_{r,k}^{(l)} - \mathbf{W}_{r,k-1}^{(l)}\|^2}{\gamma} + \sigma_0^2 + \sigma_1^2 + \sigma_1/\beta \right).$$

The overall convergence behavior is

**Theorem 3.6.** *Under appropriate conditions, by setting $\eta = \Theta(\beta) = \Theta(\gamma)$, DP version of Algorithm 1 ensures*

$$\frac{1}{R} \sum_{r} \|\nabla F(\mathbf{W}_{r-1})\|^2 \leq O\left( \frac{1}{\eta RK} + \beta + \beta^2 K^2 + \sigma_0^2 + \sigma_1^2 + \sigma_1/\beta + \sigma_2^2 + \sigma_2/\beta \right). \tag{19}$$

**Remark.** By (McMahan et al., 2018; Abadi et al., 2016), to ensure $(\epsilon, \delta)$-differential privacy, the added noise should be $\propto \frac{1}{M}$, where $M$ is the number of contributing data entries. For model updates and gradient estimator, all nodes and edges in the batch are possible to contribute, therefore, only a small noise needs to be added, but for the shared node embedding, only neighboring nodes can contribute, thus $\sigma_0$ should be much bigger than $\sigma_1$ and $\sigma_2$. Fortunately, the convergence has a much better dependence on $\sigma_0$ compared to $\sigma_1$ and $\sigma_2$.

## 4 EXPERIMENTS

We conduct experiments to assess the effectiveness, utility under differential privacy, and communication efficiency of the proposed algorithm.

**Data.** Due to regulatory constraints, real-world anti–money laundering (AML) transaction data are not publicly available. We therefore adopt the realistic simulator from Altman et al. (2023), which generates financial transaction networks by modeling agents such as banks and individuals, while injecting illicit activity through well-established laundering patterns. The reliance on synthetic data underscores the need for studying federated learning in this domain, where direct access to financial data is inherently limited.

We evaluate on two small, two medium, and two large datasets, each provided in two variants: a high illicit ratio (HI) and a low illicit ratio (LI). For example, *HI-Small* refers to the small-scale dataset with a high fraction of illicit transactions. Following Altman et al. (2023), we use a temporal train–validation–test split by transaction timestamp. To simulate federated deployment, data are partitioned across 4–32 clients. For cross-client edges, each client retains a copy of the corresponding edge features, reflecting the realistic setting of inter-bank transactions. Detailed dataset statistics are reported in Appendix D.

**Setup.** While the proposed method can be applied to message-passing GNNs with different aggregation functions as discussed in Section 3.1, in the experiments we focus on GIN (Xu et al., 2018) (a message-passing GNN with provably maximum expressive power) and PNA (Veličković et al.) (employing multiple aggregators in parallel), both augmented with edge updates, reverse messaging, Ego ID, and port numbering as described by Egressy et al. (2024). Each model consists of two message-passing layers followed by a classification head.

We compare against the following baselines: 1) Single Client (SC) that trains on each client independently, 2) FedAVG (McMahan et al., 2017) that only communicates the model weights, 3) Swift-FedGNN (Qiu et al.) that trains on global graph infrequently while only using local graphs mostly, and 4) FedGCN (Hu et al., 2022) which shares node embedding only once at the beginning.

Hyperparameters are kept consistent across methods: local seed batch size is 1k for HI datasets and 2k for LI datasets; hop-1 and hop-2 neighbor sampling sizes are both set to 100; local update steps per round are fixed to 32, unless otherwise specified. All algorithm are run for 20k iterations. Performance is measured by average F1 score across all clients.

**Primary results.** Tables 1 and 2 summarize results on HI and LI datasets. On the high-illicit (HI) datasets, our method consistently outperforms all baselines across dataset scales. In particular, Ours-PNA achieves the best overall performance—for example, on HI-Large, it improves F1 from 0.6235 (FedAVG-GIN) to 0.7114.

All methods experience performance degradation under the low-illicit (LI) setting, as expected due to stronger class imbalance. Nevertheless, our method maintains a clear advantage: Ours-PNA achieves 0.3158 F1 on LI-Large, significantly outperforming alternatives. These results suggest our approach is more robust to class imbalance, likely due to its ability to exploit structural patterns by sharing node embeddings even when illicit transactions are rare.

Table 1: Results for High Illicit Ratio Datasets.

|  | HI-Small | HI-Medium | HI-Large |
|---|---|---|---|
| SC-GIN | $0.1526 \pm 0.0157$ | $0.3572 \pm 0.0305$ | $0.2416 \pm 0.0266$ |
| SC-PNA | $0.4409 \pm 0.0294$ | $0.5305 \pm 0.0311$ | $0.5744 \pm 0.0182$ |
| FedAVG-GIN | $0.4103 \pm 0.0335$ | $0.5421 \pm 0.0273$ | $0.6235 \pm 0.0310$ |
| Swift-GIN | $0.3873 \pm 0.0306$ | $0.5689 \pm 0.0282$ | $0.6339 \pm 0.2205$ |
| FedGCN-GIN | $0.4152 \pm 0.0291$ | $0.5538 \pm 0.0310$ | $0.5817 \pm 0.0338$ |
| FedAVG-PNA | $0.5427 \pm 0.0288$ | $0.5037 \pm 0.0341$ | $0.5958 \pm 0.0299$ |
| Swift-PNA | $0.5746 \pm 0.0302$ | $0.5722 \pm 0.0258$ | $0.6144 \pm 0.0372$ |
| FedGCN-PNA | $0.5653 \pm 0.0365$ | $0.6292 \pm 0.0283$ | $0.6028 \pm 0.0324$ |
| CE-FedGNN-GIN | $0.4916 \pm 0.0218$ | $0.6024 \pm 0.0314$ | $0.6461 \pm 0.0306$ |
| CE-FedGNN-PNA | $\mathbf{0.6623 \pm 0.0273}$ | $\mathbf{0.6517 \pm 0.0322}$ | $\mathbf{0.7114 \pm 0.0251}$ |

**Communication Efficiency** We show results of varying communication interval $K$ for training CE-FedGNN-PNA in Figure 2. It is clear that skipping a lot of communication does not degrade the performance, even with $K = 1024$, our method still maintains a great advantage over other baselines shown in Table 1.

Table 2: Results for Low Illicit Ratio Datasets.

|  | LI-Small | LI-Medium | LI-Large |
|---|---|---|---|
| SC-GIN | $0.1238 \pm 0.0101$ | $0.0746 \pm 0.0153$ | $0.0080 \pm 0.0143$ |
| SC-PNA | $0.1396 \pm 0.0148$ | $0.2472 \pm 0.0200$ | $0.1262 \pm 0.0198$ |
| FedAVG-GIN | $0.0000 \pm 0.0000$ | $0.0068 \pm 0.0012$ | $0.0000 \pm 0.0000$ |
| Swift-GIN | $0.0000 \pm 0.0000$ | $0.0061 \pm 0.0016$ | $0.1294 \pm 0.0246$ |
| FedGCN-GIN | $0.1702 \pm 0.0316$ | $0.0647 \pm 0.0083$ | $0.0000 \pm 0.0000$ |
| FedAVG-PNA | $0.1556 \pm 0.0131$ | $0.2614 \pm 0.0225$ | $0.0000 \pm 0.0000$ |
| Swift-PNA | $0.1995 \pm 0.0206$ | $0.2001 \pm 0.0280$ | $0.0000 \pm 0.0000$ |
| FedGCN-PNA | $0.1726 \pm 0.0239$ | $0.1821 \pm 0.0197$ | $0.0000 \pm 0.0000$ |
| CE-FedGNN-GIN | $0.1630 \pm 0.0155$ | $0.0828 \pm 0.0127$ | $0.1917 \pm 0.0147$ |
| CE-FedGNN-PNA | $\mathbf{0.2655 \pm 0.0199}$ | $\mathbf{0.2918 \pm 0.0106}$ | $\mathbf{0.3158 \pm 0.0215}$ |

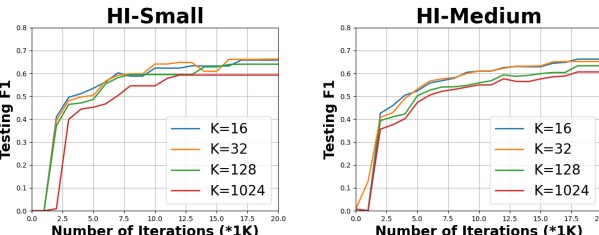

Figure 2: Ablation Study: Varying $K$

**Differential Privacy** Finally, we study the effect of differential privacy by adding Gaussian noise $\mathcal{N}(0, \sigma^2 I)$. Since the effect of noise on model/gradient are well studied, we focus on the noise on the shared embeddings, by fixing $\sigma_1 = \sigma_2 = 3 \times 10^{-3}$ and vary $\sigma_0$. We bound the Euclidean norm of embedding to 10, and set $K = 1024$ to reduce sharing frequency. The $(\epsilon, \delta)$ coefficient of differential privacy is computed using moments accountant of (Abadi et al., 2016; McMahan et al., 2018). Details are discussed in Appendix C. Results are shown in Figure 3.

We observe a clear privacy-utility tradeoff: moderate noise levels have little effect on performance, but as $\sigma_0$ increases, F1 scores degrade significantly. Interestingly, our method remains competitive with FedAVG even under stronger noise, indicating better resilience to DP perturbations.

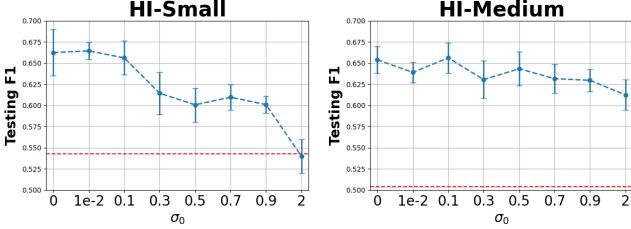

Figure 3: Performance under different $\sigma_0$. The red dashed line represents the performance of FedAVG without injected Gaussian noise.

## 5 ETHICS STATEMENT

This work introduces CE-FedGNN, a communication-efficient federated graph neural network framework for collaborative learning across distributed and privacy-sensitive graph data. All experiments are conducted on synthetic datasets simulating financial transaction networks; no proprietary, personal, or confidential real-world data were used. Our method is designed to reduce privacy risks by limiting cross-client data exchange and incorporating formal differential privacy guarantees. We acknowledge that real-world deployment in domains such as financial crime detection raises broader ethical considerations, including fairness, transparency, and potential misuse. We emphasize that further interdisciplinary research and stakeholder oversight are required before applying these techniques in operational financial or regulatory systems.

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

# A    GRAPH NEURAL NETWORK

$$\text{GCN:} \quad \mathbf{h}_v^{(l)} = \phi\left(\sum_{u \in \mathcal{N}(v) \cup \{v\}} \frac{1}{\sqrt{\tilde{d}_v \tilde{d}_u}} \mathbf{W}^{(l)} \mathbf{h}_u^{(l-1)}\right), \quad \tilde{A} = A + I, \; \tilde{d}_v = \sum_u \tilde{A}_{vu}.$$

$$\text{GIN:} \quad \mathbf{h}_v^{(l)} = \phi\left(\mathbf{W}^{(l)}\left((1+\varepsilon)\mathbf{h}_v^{(l-1)} + \sum_{u \in \mathcal{N}(v)} \mathbf{h}_u^{(l-1)}\right)\right), \quad \varepsilon^{(l)} \text{ is fixed or learnable.} \tag{20}$$

$$\text{GraphSAGE-mean:} \quad \mathbf{h}_v^{(l)} = \phi\left(\mathbf{W}^{(l)} \cdot \frac{1}{|\mathcal{N}(v) \cup \{v\}|} \sum_{u \in \mathcal{N}(v) \cup \{v\}} \mathbf{h}_u^{(l-1)}\right).$$

We denote $h_v^{(l)} = \phi(\tilde{h}_v^{(l)}) = \phi(\mathbf{W}^{(l)} \hat{h}_v^{(l)})$.

Recent advances in GNN research have explored how aggregation functions (Xu et al., 2018) and spectral perspectives (Wang & Zhang, 2022) influence expressivity, enabling models to better capture higher-order dependencies and subtle structural cues. To enhance expressive power, various techniques have been proposed, including reverse message passing, port numbering, and ego IDs (Jaume et al., 2019; Sato et al., 2019; You et al., 2021). Financial crime detection provides a concrete example where such expressivity is indispensable, as transaction networks naturally form directed multigraphs with multiple transactions between the same entities (Cardoso et al., 2022; Kanezashi et al., 2022; Weber et al., 2018; 2019; Nicholls et al., 2021). Building on these adaptations, Egressy et al. (2024) demonstrates both theoretically and empirically that expressive GNN architectures substantially improve the detection of money laundering patterns in transaction graphs. However, in realistic scenarios where data is privacy-sensitive, training such expressive GNNs in a federated setting remains a critical and largely unexplored challenge, particularly in privacy-sensitive domains such as cross-institution financial networks.

# B    ANALYSIS

We first establish the assumptions we need and then present analysis in this section. A function $f$ is said to be $C_0$-Lipschitz continuous if for all $\mathbf{x}, \mathbf{x}'$ in its domain, $\|f(\mathbf{x}) - f(\mathbf{x}')\| \leq C_0 \|\mathbf{x} - \mathbf{x}'\|$. A differentiable function $f$ is $C_1$-smooth if its gradient is Lipschitz continuous, meaning $\|\nabla f(\mathbf{x}) - \nabla f(\mathbf{x}')\| \leq C_1 \|\mathbf{x} - \mathbf{x}'\|$ for all $\mathbf{x}, \mathbf{x}'$. Let superscript $H$ denote computing forward and backward propagation using the embedding estimator $H$, while superscript $h$ denotes replacing $H$ estimator with true embeddings of sampled data. We make the following assumption throughout this paper.

**Assumption B.1.**

i) $f, g, h$ are $C_0$-Lipschitz and $C_1$-smooth over $\mathbf{W}$, and $\phi(\cdot)$ is also $C_0$-Lipschitz and $C_1$-smooth.

ii) Over a batch $B$, $\mathbb{E}\left\|\hat{\nabla}F_i^h(\mathbf{w}; B) - \hat{\nabla}F_i(\mathbf{w})\right\|^2 \leq \sigma^2$, and also $\|\hat{\nabla}F_i^H(\mathbf{w}; B)\| \leq D^2$.

iii) There exists $\mathbf{W}_*$ such that $F(\mathbf{W}) \geq F(\mathbf{W}_*) > -\infty, \forall \mathbf{W}$.

iii) $\|\mathbf{W}\|^2 \leq C_W^2$, $\|H(\cdot)\|^2 \leq C_H^2$,

## B.1    PROOF OF LEMMA 3.1

Let $p_i(v)$ denote the probability for a node $v$ to be sampled in client $i$, then we know that $p_i(v) > B_0/|\mathcal{E}_i|$ for edge based tasks, where $B_0$ is the number of seed edges, while $\mathcal{E}_i$ is the total number of edges on client $i$. Let $\tilde{h}_{r,k}^{(l)}(v)$ be the $l$-th layer's determnistic pre-activation embedding computed using model $\mathbf{W}_{r,k} := \frac{1}{N} \sum_i \mathbf{W}_{r,i,k}$ and all neighbors. Let $c(v)$ denote the hosting client of node $v$.

For the $l$-th layer with $l > 1$, we have

$$\mathbb{E}\|\tilde{H}_{r,i,k}^{(l)}(v) - \tilde{h}_{r,k}^{(l)}(v)\|^2$$

$$= p(v)\mathbb{E}\left\|(1 - \gamma)\tilde{H}_{r,i,k-1}^{(l)}(v) + \gamma\frac{1}{n_{r,i,k}(v)}\left(\mathbf{W}_{r,i,k-1}^{(l)} \cdot \sum_{u \in \mathcal{N}_{r,i,k}(v)} H_{r,i,k-1}^{(l-1)}(u)\right)\right.$$

$$\left. - \tilde{h}_{r,k-1}^{(l)}(v) + \tilde{h}_{r,k-1}^{(l)}(v) - \tilde{h}_{r,k}^{(l)}(v)\right\|^2 + (1 - p(v))\mathbb{E}\left\|\tilde{H}_{r,i,k-1}^{(l)}(v) - \tilde{h}_{i,k}^{(l)}(v)\right\|^2$$

$$= p(v)\mathbb{E}\left\|(1 - \beta)(\tilde{H}_{r,i,k-1}^{(l)}(v) - \tilde{h}_{r,k-1}^{(l)}(v))\right.$$

$$+ \gamma\left(\mathbf{W}_{r,i,k-1}^{(l)} \cdot \frac{1}{n_{r,i,k}}\left(\sum_{u \in \mathcal{N}_{r,i,k}(v)} H_{r,i,k-1}^{(l-1)}(u)\right) - \tilde{h}_{r,k-1}^{(l)}(v)\right)$$

$$\left. + \tilde{h}_{r,k-1}^{(l)}(v) - \tilde{h}_{r,k}^{(l)}(v)\right\|^2 + (1 - p(v))\mathbb{E}\left\|\tilde{H}_{r,i,k-1}^{(l)}(v) - \tilde{h}_{i,k}^{(l)}(v)\right\|^2$$

$$\leq (1 + \frac{\gamma}{4})p(v)\mathbb{E}\left\|(1 - \beta)(\tilde{H}_{r,i,k-1}^{(l)}(v) - \tilde{h}_{r,k-1}^{(l)}(v))\right.$$

$$\left. + \gamma\left(\mathbf{W}_{r,i,k-1}^{(l)} \cdot \frac{1}{n_{r,i,k}(v)}\left(\sum_{u \in \mathcal{N}_{r,i,k}(v)} H_{r,i,k-1}^{(l-1)}(u)\right) - \tilde{h}_{r,k-1}^{(l)}(v)\right)\right\|^2$$

$$+ (2 + \frac{4}{\gamma} + \frac{4}{\gamma p(v)})p(v)\left\|\tilde{h}_{r,k-1}^{(l)}(v) - \tilde{h}_{r,k}^{(l)}(v)\right\|^2 + (1 - p(v))(1 + \frac{\gamma p(v)}{4})\mathbb{E}\left\|\tilde{H}_{r,i,k-1}^{(l)}(v) - \tilde{h}_{i,k-1}^{(l)}(v)\right\|^2 \tag{21}$$

which then leads to

$$\mathbb{E}\|\tilde{H}_{r,i,k}^{(l)}(v) - \tilde{h}_{r,k}^{(l)}(v)\|^2 \leq (1 + \frac{\gamma}{4})p(v)\mathbb{E}\left\|(1 - \gamma)(\tilde{H}_{r,i,k-1}^{(l)}(v) - \tilde{h}_{r,k-1}^{(l)}(v))\right.$$

$$+ \gamma\left(\mathbf{W}_{r,i,k-1}^{(l)} \cdot \left(\frac{1}{n_{r,i,k}(v)}\sum_{u \in \mathcal{N}_{r,i,k}(v)} H_{r,i,k-1}^{(l-1)}(u) - \frac{1}{n_i(v)}\sum_{u \in \mathcal{N}_i(v)} H_{r,i,k-1}^{(l-1)}(u)\right)\right)$$

$$+ \gamma\left(\mathbf{W}_{r,i,k-1}^{(l)} \cdot \left(\frac{1}{n_i(v)}\sum_{u \in \mathcal{N}_i(v)} H_{r,c(u),k-1}^{(l-1)}(u) - \tilde{h}_{r,k-1}^{(l)}(v)\right)\right)\Bigg\|^2$$

$$+ \frac{5}{\gamma}\left\|\tilde{h}_{r,k-1}^{(l)}(v) - \tilde{h}_{r,k}^{(l)}(v)\right\|^2 + (1 - p(v))(1 + \frac{\gamma p(v)}{4})\mathbb{E}\left\|\tilde{H}_{r,i,k-1}^{(l)}(v) - \tilde{h}_{i,k-1}^{(l)}(v)\right\|^2. \tag{22}$$

Then noting

$$\mathbb{E}_{r,k-1}\left(\frac{1}{n_{r,i,k}(v)}\sum_{u \in \mathcal{N}_{r,i,k}(v)} H_{r,i,k-1}^{(l-1)}(u) - \frac{1}{n_i(v)}\sum_{u \in \mathcal{N}_i(v)} H_{r,c(u),k-1}^{(l-1)}(u)\right) = 0, \tag{23}$$

we have

$$
\mathbb{E}\|\tilde{H}_{r,i,k}^{(l)}(v) - \tilde{h}_{r,k}^{(l)}(v)\|^2
$$

$$
\leq (1 + \frac{\gamma}{4})p(v)\mathbb{E}\left\|(1-\gamma)(\tilde{H}_{r,i,k-1}^{(l)}(v) - \tilde{h}_{r,k-1}^{(l)}(v))\right.
$$

$$
\left. + \gamma\left(\mathbf{W}_{r,i,k-1}^{(l)} \cdot \frac{1}{n_i(v)}\sum_{u \in \mathcal{N}(v)} H_{r,i,k-1}^{(l-1)}(u) - \tilde{h}_{r,k-1}^{(l)}(v)\right)\right\|^2
$$

$$
+ \gamma^2 G^2 + \frac{5}{\gamma}\left\|\tilde{h}_{r,k-1}^{(l)}(v) - \tilde{h}_{r,k}^{(l)}(v)\right\|^2 + (1-p(v))(1 + \frac{\gamma p(v)}{4})\mathbb{E}\left\|\tilde{H}_{r,i,k-1}^{(l)}(v) - \tilde{h}_{i,k-1}^{(l)}(v)\right\|^2
$$

$$
\leq (1 + \frac{\gamma}{4})^2(1-\gamma)^2 p(v)\mathbb{E}\left\|\tilde{H}_{r,i,k-1}^{(l)}(u) - \tilde{h}_{i,k-1}^{(l)}(u)\right\|^2
$$

$$
+ (1 + \frac{4}{\gamma})\gamma^2\|\left(\mathbf{W}_{r,i,k-1}^{(l)} \cdot \frac{1}{n_i(v)}(\sum_{u \in \mathcal{N}(v)} H_{r,i,k-1}^{(l-1)}(u) - \tilde{h}_{r,k-1}^{(l)}(v))\right)\|^2 + \gamma^2 G^2
$$

$$
+ \frac{5}{\gamma}\left\|\tilde{h}_{r,k-1}^{(l)}(v) - \tilde{h}_{r,k}^{(l)}(v)\right\|^2 + (1-p(v))(1 + \frac{\gamma p(v)}{4})\mathbb{E}\left\|\tilde{H}_{r,i,k-1}^{(l)}(v) - \tilde{h}_{i,k-1}^{(l)}(v)\right\|^2
$$

$$
\leq (1 - \frac{\gamma p(v)}{4})\mathbb{E}\left\|\tilde{H}_{r,i,k-1}^{(l)}(u) - \tilde{h}_{i,k-1}^{(l)}(u)\right\|^2 + 8\gamma C_W^2 \frac{1}{n_i(v)}\sum_{u \in \mathcal{N}(v)}\|\tilde{H}_{r,i,k-1}^{(l-1)}(u) - \tilde{h}_{r,k-1}^{(l-1)}(u)\|^2)
$$

$$
+ \gamma^2 C_W^2 C_H^2 + \frac{8}{\gamma}C^2\|\mathbf{W}_{r,k} - \mathbf{W}_{r,k-1}\|^2.
\tag{24}
$$

where $C$ is a Lipschitz constant of $\tilde{h}_{r,k}^{(l)}$ over $\mathbf{W}$, depending on the constants in Assumption B.1. Rearranging the terms, we have

$$
\mathbb{E}\left\|\tilde{H}_{r,i,k-1}^{(l)}(v) - \tilde{h}_{i,k-1}^{(l)}(v)\right\|^2 \leq \frac{4\left(\mathbb{E}\left\|\tilde{H}_{r,i,k-1}^{(l)}(v) - \tilde{h}_{i,k-1}^{(l)}(v)\right\|^2 - \mathbb{E}\left\|\tilde{H}_{r,i,k}^{(l)}(v) - \tilde{h}_{i,k}^{(l)}(v)\right\|^2\right)}{\gamma p(v)}
$$

$$
+ \frac{32C_W^2}{p(v)n_i(v)}\sum_{u \in \mathcal{N}(v)}\|\tilde{H}_{r,i,k-1}^{(l-1)}(u) - \tilde{h}_{r,k-1}^{(l-1)}(u)\|^2) + \frac{4}{p(v)}\gamma C_W^2 C_H^2 + \frac{32}{\gamma^2 p(v)}C^2\|\mathbf{W}_{r,k} - \mathbf{W}_{r,k-1}\|^2.
\tag{25}
$$

Note that $\tilde{H}_{r,i,0}^{(l)} = \tilde{H}_{r-1,i,K}^{(l)}$, and $\tilde{h}_{r,0}^{(l)} = \tilde{h}_{r-1,K}^{(l)}$. Let $p := \min_{u \in \mathcal{V}} p(u)$. Taking the telescoping sum, we have

$$
\frac{1}{RK}\sum_r\sum_k \mathbb{E}\left\|\tilde{H}_{r,i,k-1}^{(l)}(v) - \tilde{h}_{i,k-1}^{(l)}(v)\right\|^2 \leq O\left(\frac{4(L+1)C^2}{\gamma p RK}\right.
$$

$$
\left. + \frac{4}{p}\gamma(L+1)G^2 + \frac{32}{\gamma^2 p}(L+1)C^2\|\mathbf{W}_{r,k} - \mathbf{W}_{r,k-1}\|^2\right).
\tag{26}
$$

And thus using Lipschitz of $\phi(\cdot)$, we obtain

$$
\frac{1}{RK}\sum_r\sum_k \mathbb{E}\left\|H_{r,i,k-1}^{(l)}(v) - h_{i,k-1}^{(l)}(v)\right\|^2 \leq O\left(\frac{4(L+1)C^2}{\gamma p RK}\right.
$$

$$
\left. + \frac{4}{p}\gamma(L+1)G^2 + \frac{32}{\gamma^2 p}(L+1)C^2\|\mathbf{W}_{r,k} - \mathbf{W}_{r,k-1}\|^2\right).
\tag{27}
$$

## B.2 PROOF OF LEMMA 3.2

Let superscript $H$ denote computing forward and backward propagation using the embedding estimator $H$, while superscript $h$ denotes replacing $H$ estimator with true embeddings of sampled data.

By the update rule of the gradient estimator, we have that

$$\|\bar{G}_{r,k} - \nabla F(\mathbf{W}_{r,k})\|^2$$

$$\leq (1+\frac{\beta}{4})\left\|(1-\beta)\bar{G}_{r,k-1} + \beta\frac{1}{N}\sum_{i=1}^{N}\left(\hat{\nabla}F_i(\mathbf{W}_{r,i,k-1};B_{r,i,k})\right) - \nabla F(\mathbf{W}_{r,k-1})\right\|^2$$

$$+ (1+\frac{4}{\beta})C^2\|\mathbf{W}_{r,k-1} - \mathbf{W}_{r,k}\|^2$$

$$= (1+\frac{\beta}{4})\Big\|(1-\beta)(\bar{G}_{r,k-1} - \nabla F(\mathbf{W}_{r,k-1}))$$

$$+ \beta\frac{1}{N}\sum_{i=1}^{N}\left(\hat{\nabla}F_i^{H}(\mathbf{W}_{r,i,k-1};B_{r,i,k}) - \hat{\nabla}F_i^{h}(\mathbf{W}_{r,i,k-1};B_{r,i,k})\right)$$

$$+ \beta\frac{1}{N}\sum_{i=1}^{N}\left(\hat{\nabla}F_i^{h}(\mathbf{W}_{r,i,k-1};B_{r,i,k}) - \nabla F(\mathbf{W}_{r,k-1})\right)\Big\|^2 + (1+\frac{4}{\beta})C^2\|\mathbf{W}_{r,k-1} - \mathbf{W}_{r,k}\|^2$$

$$= (1+\frac{\beta}{4})\Big\|(1-\beta)(\bar{G}_{r,k-1} - \nabla F(\mathbf{W}_{r,k-1}))$$

$$+ \beta\frac{1}{N}\sum_{i=1}^{N}\left(\hat{\nabla}F_i^{h}(\mathbf{W}_{r,i,k-1};B_{r,i,k}) - \nabla F(\mathbf{W}_{r,k-1})\right)\Big\|^2$$

$$+ (1+\frac{4}{\beta})\beta^2\frac{1}{N}\sum_{i=1}^{N}\left\|\left(\hat{\nabla}F_i^{H}(\mathbf{W}_{r,i,k-1};B_{r,i,k}) - \hat{\nabla}F_i^{h}(\mathbf{W}_{r,i,k-1};B_{r,i,k})\right)\right\|^2$$

$$+ (1+\frac{4}{\beta})C^2\|\mathbf{W}_{r,k-1} - \mathbf{W}_{r,k}\|^2$$

$$\leq (1+\frac{\beta}{4})^2\Big\|(1-\beta)(\bar{G}_{r,k-1} - \nabla F(\mathbf{W}_{r,k-1}))$$

$$+ \beta\frac{1}{N}\sum_{i=1}^{N}\left(\hat{\nabla}F_i^{h}(\mathbf{W}_{r,k-1};B_{r,i,k}) - \nabla F(\mathbf{W}_{r,k-1})\right)\Big\|^2 + (1+\frac{4}{\beta})\beta^2\frac{1}{N}\sum_{i=1}^{N}\|\mathbf{W}_{r,k-1} - \mathbf{W}_{r,i,k-1}\|^2$$

$$+ (1+\frac{4}{\beta})\beta^2\frac{1}{N}\sum_{i=1}^{N}\left\|\left(\hat{\nabla}F_i^{H}(\mathbf{W}_{r,i,k-1};B_{r,i,k}) - \hat{\nabla}F_i^{h}(\mathbf{W}_{r,i,k-1};B_{r,i,k})\right)\right\|^2$$

$$+ (1+\frac{4}{\beta})C^2\|\mathbf{W}_{r,k-1} - \mathbf{W}_{r,k}\|^2$$

$$\leq (1-\frac{\beta}{4})\|\bar{G}_{r,k-1} - \nabla F(\mathbf{W}_{r,k-1})\|^2 + \frac{\beta^2\sigma^2}{N} + C^2 5\beta\eta^2 K^2 D^2$$

$$+ 5\beta\frac{1}{N}\sum_{i=1}^{N}\|\hat{\nabla}F_i^{H'}(\mathbf{W}_{r,i,k-1};B_{r,i,k}) - \hat{\nabla}F_i^{h}(\mathbf{W}_{r,i,k-1};B_{r,i,k})\|^2 + \beta^3 K^2 C_H^2$$

$$+ \frac{5C^2\eta^2}{\beta}\|\bar{G}_{r,k-1}\|^2,$$

$$(28)$$

where superscript $H'$ denoting plugging all $H$ estimators at iteration $(r,i,k)$. For remote nodes, the difference between current estimator and previous estimator is bounded by $\beta^2 K^2 C_H^2$. And

$$\|\hat{\nabla}F_i^{H}(\mathbf{W}_{r,i,k-1};B_{r,i,k}) - \hat{\nabla}F_i^{h}(\mathbf{W}_{r,i,k-1};B_{r,i,k})\|^2 \leq \sum_{u\in B_{r,i,k}} C^2 m(u,l)\|\tilde{H}^{(l)r,i,k}(u) - \tilde{h}_{r,i,k}^{(l)}(u)\|^2,$$

$$(29)$$

with $m(u,l)$ is the weight for node $u$ at layer $l$ satisfying $\sum_{u\in B_{r,i,k}} m(u,l) = 1$.

Rearranging terms, applying telescoping sum and plugging Lemma 3.5,

$$
\begin{aligned}
\frac{1}{RK} \sum_r \sum_k \|(\bar{G}_{r,k-1} - \nabla F(\mathbf{W}_{r,k-1}))\|^2 \leq O\Bigg( & \frac{1}{\beta RK} + \frac{1}{\gamma p RK} \\
& + \frac{\beta}{N} + \frac{\gamma}{p} + \beta^2 K^2 + \frac{8C^2\eta^2}{\beta^2}\|\bar{G}_{r,k-1}\|^2 \Bigg).
\end{aligned}
\tag{30}
$$

With $\eta = O(\beta)$, we have

$$
\begin{aligned}
\frac{1}{RK} \sum_r \sum_k \|(\bar{G}_{r,k-1} - \nabla F(\mathbf{W}_{r,k-1}))\|^2 \leq O\Bigg( & \frac{1}{\beta RK} + \frac{1}{\gamma p RK} \\
& + \frac{\beta}{N} + \frac{\gamma}{p} + \beta^2 K^2 + \frac{\gamma^2 K^2}{p} + \frac{\eta^2}{\beta^2}\|\nabla F(\mathbf{W}_{r,k-1})\|^2 \Bigg).
\end{aligned}
\tag{31}
$$

### B.3 PROOF OF THEOREM 3.3

Using $C_1$-smooth of $F$, we have

$$
\begin{aligned}
F(\mathbf{W}_r) \leq{} & F(\mathbf{W}_{r-1}) + \langle \nabla F(\mathbf{W}_{r-1}), \mathbf{W}_r - \mathbf{W}_{r-1} \rangle + \frac{C_1}{2}\|\mathbf{W}_r - \mathbf{W}_{r-1}\|^2 \\
={} & F(\mathbf{W}_{r-1}) - \langle \nabla F(\mathbf{W}_{r-1}), \eta\frac{1}{N}\sum_i\sum_k G_{r,i,k}(\mathbf{W}_{r,i,k})\rangle + \frac{C_1}{2}\|\mathbf{W}_r - \mathbf{W}_{r-1}\|^2 \\
={} & F(\mathbf{W}_{r-1}) - \langle \nabla F(\mathbf{W}_{r-1}), \eta\frac{1}{N}\sum_i\sum_k \hat{\nabla} F_i(\mathbf{W}_{r-1})\rangle + \frac{C_1}{2}\|\mathbf{W}_r - \mathbf{W}_{r-1}\|^2 \\
& - \langle \nabla F(\mathbf{W}_{r-1}), \eta\frac{1}{N}\sum_i\sum_k G_{r,i,k} - \eta\frac{1}{N}\sum_i\sum_k \hat{\nabla} F_i(\mathbf{W}_{r-1})\rangle \\
={} & F(\mathbf{W}_{r-1}) - \eta K\|F(\mathbf{W}_{r-1})\|^2 + \frac{\eta}{2}K\|\nabla F(\mathbf{W}_{r-1})\|^2 \\
& + \frac{\eta}{K}\left\|\frac{1}{N}\sum_i\sum_k G_{r,i,k} - \frac{1}{N}\sum_i\sum_k \hat{\nabla} F_i(\mathbf{W}_{r-1})\right\|^2 \\
& + \frac{\eta K}{NK}\sum_i\sum_k \|\mathbf{W}_{r-1} - \mathbf{W}_{r,i,k}\|^2 + \frac{C_1}{2}\eta^2\left\|\frac{1}{N}\sum_i\sum_k G_{r,i,k}\right\|^2.
\end{aligned}
\tag{32}
$$

Thus, plugging Lemma 3.2, we have

$$
\begin{aligned}
& \frac{1}{R}\sum_r \mathbb{E}\|\nabla F(\mathbf{W}_{r-1})\|^2 \\
& \leq O\Bigg( \frac{F(\mathbf{W}_0) - F(\mathbf{W}_*)}{\eta RK} + \frac{1}{RK}\sum_r\sum_k \|(\bar{G}_{r,k-1} - \nabla F(\mathbf{W}_{r,k-1}))\|^2 + \eta^2 K^2 D^2 \Bigg) \\
& \leq O(\frac{1}{\eta RK} + \frac{1}{\beta RK} + \frac{1}{\gamma p RK} + \frac{\beta}{N} + \frac{\gamma}{p} + \beta^2 K^2).
\end{aligned}
\tag{33}
$$

## C  ANALYSIS UNDER DIFFERENTIAL PRIVACY

The analysis of Lemma 3.4 is based on Lemma 3.1. Within a round $r$, we have

$$
\mathbb{E}\left\|\tilde{H}_{r,i,k-1}^{(l)}(v) - \tilde{h}_{i,k-1}^{(l)}(v)\right\|^2 \leq \frac{4\left(\mathbb{E}\left\|\tilde{H}_{r,i,k-1}^{(l)}(v) - \tilde{h}_{i,k-1}^{(l)}(v)\right\|^2 - \mathbb{E}\left\|\tilde{H}_{r,i,k}^{(l)}(v) - \tilde{h}_{i,k}^{(l)}(v)\right\|^2\right)}{\beta p(v)}
$$
$$
+ \frac{32C_W^2}{p(v)n_i(v)} \sum_{u \in \mathcal{N}(v)} \|\tilde{H}_{r,i,k-1}^{(l-1)}(u) - \tilde{h}_{r,k-1}^{(l-1)}(u)\|^2) + \frac{4}{p(v)}\beta C_W^2 C_H^2 + \frac{32}{\beta^2 p(v)} C^2 \|\mathbf{W}_{r,k}^{(l)} - \mathbf{W}_{r,k-1}^{(l)}\|^2.
$$
(34)

Due to the noise to $\mathbf{W}$ at the global communication round, we have

$$
\mathbb{E}\|\tilde{H}_{r,i,K}^{(l)}(v) - \tilde{h}_{r,i,K}^{(l)}(v)\|^2
$$
$$
= \mathbb{E}\|\tilde{H}_{r,i,K}^{(l)}(v) - \tilde{h}_{r+1,i,0}^{(l)}(v) + \tilde{h}_{r+1,i,0}^{(l)}(v) - \tilde{h}_{i,K}^{(l)}(v)\|^2
$$
(35)
$$
\leq \mathbb{E}\|\tilde{H}_{r,i,K}^{(l)}(v) - \tilde{h}_{r+1,i,0}^{(l)}(v)\|^2 + C_H\sigma_1 + \sigma_1^2.
$$

As a result, taking the telescoping sum yields

$$
\frac{1}{RK} \sum_r \sum_k \mathbb{E}\left\|\tilde{H}_{r,i,k-1}^{(l)}(v) - \tilde{h}_{i,k-1}^{(l)}(v)\right\|^2
$$
$$
\leq O\left(\frac{1}{\gamma RK} + \gamma + \gamma^2 K^2 + \frac{\|\mathbf{W}_{r,k} - \mathbf{W}_{r,k-1}\|^2}{\gamma^2} + \frac{\sigma_1 + \sigma_1^2}{\gamma}\right),
$$
(36)

which concludes Lemma 3.4.

Similarly, based on the analysis of Lemma 3.2, we can have Lemma 3.5. Besides the noise in $\mathbf{W}$ as we handled above, it also depends on noise to $H$, which would add a $\sigma_0^2$ to (29). Also, gradient estimator $G$ is added a noise of $\mathcal{G}(0, \sigma_2^2)$. Therefore,

$$
\frac{1}{RK} \sum_r \sum_k \|(\bar{G}_{r,k-1} - \nabla F(\mathbf{W}_{r,k-1}))\|^2 \leq
$$
$$
O\left(\frac{1}{\beta RK} + \frac{1}{\gamma RK} + \beta + \gamma + \beta^2 K^2 + \frac{\eta^2}{\beta^2}\|\nabla F(\mathbf{W}_{r,k-1})\|^2 + \sigma_0^2 + \sigma_1^2 + \sigma_1/\beta + \sigma_2^2 + \sigma_2/\beta\right)
$$
(37)

Then Theorem 3.6 follows as

$$
\frac{1}{R} \sum_r \mathbb{E}\|\nabla F(\mathbf{W}_{r-1})\|^2 \leq O\left(\frac{1}{\eta RK} + \beta + \beta^2 K^2 + \sigma_0^2 + \sigma_1^2 + \sigma_1/\beta + \sigma_2^2 + \sigma_2/\beta\right).
$$
(38)

**Noise Level in Differential Privacy**  We focus on the differential privacy guarantee of shared embeddings. Following the common practice of clipping McMahan et al. (2018), we clip the Euclidean norm each shared embedding to be less than 10 by and only share node embeddings when there are at least 20 neighbor data points have been involved to update its embedding. Then we set $\delta$ in $(\epsilon, \delta)$ to be 0.01, thus the corresponding upper bound of $\epsilon$ regarding to $\sigma_0$ is as follows in the 20 communication rounds (each round has 1024 iterations): $\sigma_0 = 1e - 2 : \epsilon \leq 50004$, $\sigma_0 = 0.1 : \epsilon \leq 505$, $\sigma_0 = 0.3 : \epsilon \leq 61$, $\sigma_0 = 0.5 : \epsilon \leq 25$, $\sigma_0 = 0.7 : \epsilon \leq 15$, $\sigma_0 = 0.9 : \epsilon \leq 11$, $\sigma_0 = 2 : \epsilon \leq 5$.

## D  DATA STATISTICS

Data statistics is summarized in Table 3. L/S denotes the ratio of the number of data on the largest client over that on the smallest client.

For the two small datasets, training days are 0-7, validation day is 8, and testing days are 9-13. For the two medium datasets, training days are 0-13, validation day is 14, and testing days are 15-26. For the two large datasets, training days are 0-94, validation day is 95, and testing days are 96-162.

Table 3: Statistics of the Datasets

|           | # of Edges   | # of Nodes | # of Clients | L/S  |
|-----------|--------------|------------|--------------|------|
| HI-Small  | 5,078,345    | 515,088    | 4            | 1.37 |
| HI-Medium | 31,898,238   | 2,077,023  | 6            | 1.52 |
| HI-Large  | 179,702,229  | 2,116,168  | 32           | 3.47 |
| LI-Small  | 6,924,049    | 705,907    | 4            | 1.28 |
| LI-Medium | 31,251,483   | 2,032,095  | 6            | 1.55 |
| LI-Large  | 179,702,229  | 2,070,980  | 16           | 3.15 |

Figure 4: AIA attack under different hop-1 and hop-2 size. Legends are in the form of (hop-1, hop-2).

## E  EXPERIMENTS ON ATTRIBUTE INFERENCE ATTACK

In this subsection, we focus on the HI-Small dataset to study attribute inference attacks (AIA) on our methods. We concentrate on AIA because it represents a realistic privacy threat in federated GNNs. Unlike MIA or GSR Zhang et al. (2024), AIA better models adversaries in financial networks who may access partial embeddings and aim to recover sensitive node attributes. We assume the attacker knows a node's membership in the targeted bank, along with a subset of nodes and connections in other banks. Leveraging this information together with shared embeddings, the attacker attempts to reconstruct the remaining unknown features. Reconstruction quality is evaluated using the following loss function.

$$\arg\min_{x'} \|h^{(L)}(x'; B \cup \{x\}) - h^{(L)}(x; B \cup \{x\})\|^2. \tag{39}$$

We then measure the reconstruction quality using the Mean Squared Error (MSE) between the true features $x$ and the reconstructed features $x'$.

The results, shown in Figure 4, indicate that embeddings computed with more neighbors are harder to reconstruct. Additionally, we benchmarked the average distance to the top 1% nearest neighbors in the dataset. The attribute inference attack (AIA) fails to reconstruct raw features within this 1% threshold, verifying that sharing aggregated embeddings provides strong robustness against privacy attacks.

## F  USING EXTERNAL NEIGHBORS FOR NODE EMBEDDING AGGREGATION

Let the subscript $r, i, k$ denote round $r$ at machine $i$ and iteration $k$. The forward pass for a node using a moving average is defined as:

$$\tilde{H}^{(l)}_{r,i,k}(v) = (1-\gamma)\tilde{H}^{(l)}_{r,i,k-1}(v) + \gamma \frac{1}{n_{r,i,k}(v)} \mathbf{W}^{(l,n)}_{r,i,k-1} \cdot \Big( \sum_{u \in \mathcal{N}_{r,i,k}(v)} H^{(l-1)}_{r,i,k}(u) + \boxed{\sum_{u \in \mathcal{N}'_{r,i,K}(v,l)} H^{(l-1)}_{r-1,c(u),K}(u)} \Big),$$

$$H^{(l)}_{r,i,k}(v) = \phi(\tilde{H}^{(l)}_{r,i,k}(v)),$$

$$\tag{40}$$

Here, $\mathcal{N}_{r,i,k}(v)$ denotes the local neighbors in the current batch, while $\mathcal{N}'_{r,i,K}(v,l)$ corresponds to neighbors from other clients whose embeddings were previously shared. Note that $\mathcal{N}'_{r,i,K}(v,l)$

depends on $l$ since that we define a node's neighbors varying with respect to layers to utilize only high level embeddings from remote client to save communication time and reduce privack risk. The embeddings $H^{(l-1)}r-1, c(u), K(u)$ are retrieved from other clients since $u$ is a remote node. And $n_{r,i,k}(v) = |\mathcal{N}_{r,i,k}(v) \cup \mathcal{N}'r, i, K(v,l)|$. Input embeddings are set as $H^{(0)}(\cdot) = h^{(0)}(\cdot)$.

In addition to sharing $W, G, H$ as described in the main text, we also share portions of the chain rule with other clients to enable gradient computation. These shared components can similarly use a moving average estimator to control variance, though in practice sharing them is optional.

