# OpenReview forum: "Provably Communication-Efficient Federated Graph Neural Network"
_ICLR.cc/2026/Conference — Submitted to ICLR 2026_

### Official Review · Reviewer_e5Xn · 2025-10-31

**Soundness:** 1
**Presentation:** 1
**Contribution:** 2
**Rating:** 2
**Confidence:** 3

**Summary:**

This paper proposes CE-FedGNN, a federated graph neural network (GNN) framework that integrates cross-client edges while minimizing communication overhead. By incorporating differential privacy, CE-FedGNN provides formal privacy guarantees. Experimental evaluation on a synthetic interbank anti-money laundering task demonstrates that CE-FedGNN consistently outperforms selected baselines.

**Strengths:**

1. The paper addresses an important and timely problem, federated learning on graph-structured data, which has broad potential applications in domains such as finance, healthcare, and social networks.

**Weaknesses:**

1. The notation is unclear and lacks rigorous definitions, making the algorithmic details difficult to follow.
2. The related work discussion is incomplete. Several key studies on cross-client edge modeling in federated GNNs are missing, raising novelty concerns.
3. The experimental section is weak. Only one dataset is used, and there are no communication overhead comparisons with relevant baselines.

**Questions:**

Based on the review above and some other issues, the reviewers have the following questions or comments.

1. The current notations are inconsistent and not well-defined. This makes it difficult to fully understand the workflow and theoretical design of the proposed algorithm. Please revise and clarify all variables, symbols, and equations.
2. The paper should include discussions on representative studies that explicitly handle cross-client edge modeling, such as FedSage [1] and FED-PUB [2]. These methods should also be included as experimental baselines to ensure a fair comparison.
3. The experiments rely on only one synthetic dataset. Additional experiments on datasets from other domains (e.g., citation networks, social graphs) would help demonstrate the generalizability of CE-FedGNN.
4. Since one of CE-FedGNN’s main claims is reduced communication cost, quantitative comparisons of communication overhead with other federated GNN methods are needed to substantiate this claim.
5. The paper mentions convergence rate and communication complexity but provides no formal proof or empirical validation. The authors should either include theoretical justification or remove these claims.
6. The paper mentions a “biased gradients” challenge on Page 4, but does not clearly explain its origin or impact. Please clarify whether this is a federated learning-specific problem or a general deep learning issue.

[1] Ke Zhang, Carl Yang, Xiaoxiao Li, Lichao Sun, Siu-Ming Yiu: Subgraph Federated Learning with Missing Neighbor Generation. NeurIPS 2021: 6671-6682

[2] Jinheon Baek, Wonyong Jeong, Jiongdao Jin, Jaehong Yoon, Sung Ju Hwang: Personalized Subgraph Federated Learning. ICML 2023: 1396-1415

---

### Official Review · Reviewer_VT7c · 2025-11-01

**Soundness:** 2
**Presentation:** 2
**Contribution:** 1
**Rating:** 2
**Confidence:** 4

**Summary:**

This paper addresses the communication-accuracy-privacy trade-off in federated graph neural networks and proposes CE-FedGNN. The framework uses moving-average estimation for node embeddings/gradients and infrequently transmits high-level embeddings to reduce overhead and privacy leakage. It provides theoretical guarantees to a stationary point, communication complexity and formal differential privacy via Gaussian noise injection. Experiments on synthetic cross-bank anti-money laundering (AML) data show CE-FedGNN outperforms baselines.

**Strengths:**

1. Focuses on a critical, unresolved issue in FedGNNs, existing methods either ignore cross-client edges or have high communication costs, aligning with real needs.
2. Provides formal analysis for convergence, communication complexity, and DP, which is rare in FedGNN works and strengthens the method’s credibility.
3. Simulates realistic federated settings (edge attributes shared, node attributes private) and tests on AML tasks (a privacy-sensitive domain), making the work relevant to real-world applications.

**Weaknesses:**

1. Relies Exclusively on Synthetic, Non-Public Data: All experiments use synthetic AML data generated by a simulator. There is no validation on public FedGNN benchmarks. This means CE-FedGNN’s performance on non-AML tasks or real-world graphs is unproven. Additionally, key details for data generation are not shared, harming reproducibility.
2. Theoretical Assumptions Conflict with Practical GNN Use: The theory requires activation functions to be "smooth", but mainstream GNN activations (e.g., ReLU) are not smooth. This creates an unavoidable issue: either CE-FedGNN cannot be used with most practical GNNs, or the entire convergence analysis is invalid.
3. Core Designs Lack GNN-Specific Innovation: The two key ideas are adaptations of existing federated learning concepts. For example, moving averages are used in FL methods like FedProx to control variance. The paper does not add unique designs tailored to GNNs.
4. Differential Privacy Analysis Is Incomplete: The paper only analyzes noise effects in a single communication round, but FedGNN training requires multiple rounds and noise accumulation across rounds (which impacts both DP budget and accuracy) is not studied. It also fails to provide a clear quantitative relationship between noise level, DP budget (ϵ), and model accuracy, making the DP feature hard to use in practice.
5. Baseline Comparisons Are Inadequate: The paper omits recent FedGNN baselines focused on privacy. It also lacks a "Local GNN" baseline that ignores cross-client edges entirely, so the actual benefit of CE-FedGNN’s cross-client edge handling cannot be quantified.

**Questions:**

1. Can CE-FedGNN be adjusted to work with non-smooth activations while keeping its theoretical guarantees?
2. Would CE-FedGNN maintain its performance advantage when tested on public FedGNN datasets (not just synthetic AML data)?

---

### Official Review · Reviewer_LLUi · 2025-11-01

**Soundness:** 3
**Presentation:** 3
**Contribution:** 3
**Rating:** 6
**Confidence:** 4

**Summary:**

CE-FedGNN achieves a balance between accuracy, communication efficiency, and privacy for federated GNNs. It uses moving-average embeddings/gradients and low-frequency high-level embedding transmission. Gaussian noise injection ensures differential privacy, and experiments on anti-money laundering data show it outperforms baselines even with privacy guarantees.

**Strengths:**

• Communication efficiency: Avoids per-iteration embedding sharing, cutting complexity;
• Captures cross-client edges via 1-hop shared embeddings, outperforming methods that ignore global structure.
• Privacy guarantee: Formal DP via Gaussian noise, with >95% accuracy retained under moderate noise.

**Weaknesses:**

• Relies on cross-client edge attribute sharing (e.g., inter-bank transaction details), which may not hold in all scenarios.
• Requires manual tuning of DP noise intensity without automation.
• While tested with up to 32 clients, scalability to hundreds or thousands of clients typical in large-scale FL systems is not evaluated.

**Questions:**

AML is an essential task in financial risk control, simulation dataset may not be sufficient to show the effectiveness in reality.

---

### Official Review · Reviewer_xVhn · 2025-11-01

**Soundness:** 3
**Presentation:** 2
**Contribution:** 2
**Rating:** 4
**Confidence:** 2

**Summary:**

This paper introduces CE-FedGNN, a communication-efficient federated learning framework for Graph Neural Networks (GNNs) operating on graphs distributed across multiple clients with inter-client edges. The core challenge is that accurate GNN training requires information from a node's neighbors, which may reside on other clients, but frequent exchange of embeddings incurs prohibitive communication costs and privacy risks. CE-FedGNN addresses this by using moving-average estimators for both node embeddings and gradients. Clients infrequently share only these aggregated, high-level embeddings instead of raw data or per-iteration updates. The authors provide a convergence analysis for their non-convex optimization problem, proving a rate of \(O(1/\sqrt{T})\) to a stationary point with a communication complexity of \(O(T^{3/4})\). They also extend the framework with Gaussian noise injection for differential privacy and analyze its convergence. Experiments on a synthetic anti-money laundering task demonstrate the method's effectiveness, communication efficiency, and utility under DP constraints.

**Strengths:**

1.  Well-Motivated and Important Problem: The paper tackles a critical and realistic problem at the intersection of federated learning and graph learning. The scenario of a graph distributed across clients with connecting edges (e.g., cross-bank transactions) is common in practice but poorly handled by standard FL methods. The motivation is clear and compelling.
2.  Novel Algorithmic Design: The use of moving-average estimators for both embeddings (H) and gradients (G) is a clever and novel core contribution. This design elegantly decouples the need for frequent communication from the training process, allowing clients to use slightly stale but low-variance estimates of their neighbors' states. The insight to apply this specifically to nodes (not edges) is practical, as nodes are typically fewer than edges.
3.  Significant Theoretical Contributions: Providing convergence guarantees for a federated GNN algorithm is non-trivial due to the multi-layer compositional structure and cross-client dependencies. The paper makes a substantial theoretical contribution by proving convergence rates and communication complexity for their non-convex problem. Extending this analysis to the differentially private version is another key strength.
4.  Comprehensive Experimental Evaluation: The experiments are well-designed, using a realistic synthetic financial dataset (anti-money laundering) that mirrors the paper's motivating application. The comparison against relevant baselines (Single Client, FedAvg, Swift-FedGNN, FedGCN) across different dataset scales and illicit transaction ratios (High/Low) provides strong empirical support. The ablation studies on communication intervals (K) and DP noise levels are particularly informative.

**Weaknesses:**

1.  Dependence on Synthetic Data: While justified by the unavailability of real financial data, the exclusive reliance on a single synthetic dataset (from Altman et al., 2023) is a limitation. The generalizability of the results to other graph types (e.g., social networks, citation networks) and real-world distributions remains unproven. Performance on standard graph benchmark tasks would strengthen the claims.
2.  Simplified Privacy Analysis and Threat Model: The DP analysis focuses on adding Gaussian noise to shared embeddings. However, the overall privacy guarantee is somewhat nuanced. The privacy accounting (in Appendix C) considers the embedding sharing in isolation. A more comprehensive analysis would consider the end-to-end privacy loss accounting for all shared information (models, gradients, embeddings) together. Furthermore, the threat model primarily considers attribute inference attacks (AIA); a discussion of other potential attacks like membership inference or graph reconstruction attacks would provide a more complete security picture.
3.  Assumptions in Theoretical Analysis: The convergence proofs rely on several standard but strong assumptions (e.g., Lipschitz continuity and smoothness of functions, bounded gradients, existence of a minimum). The practical impact of these assumptions, especially for complex GNN architectures and real-world graph data which may not perfectly satisfy them, is not deeply discussed.
4.  Limited Discussion of System Overheads: While communication rounds are reduced, the local computation and memory overhead of maintaining moving-average estimators (H and G) for all relevant nodes is not thoroughly analyzed. For very large graphs, this storage cost per client could be non-negligible and might affect the practical scalability.

**Questions:**

1.  How does the memory and computational overhead of maintaining the moving-average estimators H and G scale with graph size and number of clients? Could this become a bottleneck for massive graphs?
2.  Beyond the synthetic financial transaction graph, have you tested CE-FedGNN on other types of graph data (e.g., social networks, molecular graphs) to demonstrate its general applicability? What adaptations, if any, are needed?
3.  The privacy analysis focuses on noise added to embeddings. How would you perform a comprehensive privacy accounting that composes the privacy loss from sharing model parameters, gradients, and embeddings throughout the training process?

---

### Meta-Review · Area_Chair_mhRH · 2026-01-07

**Summary:**

The paper proposes CE-FedGNN, a framework for federated graph learning that utilizes moving-average estimators and differential privacy to address communication and privacy constraints. While the motivation is well-received, the reviewers identified significant deficiencies in the experimental validation, theoretical assumptions, and comparison with the state-of-the-art.

**Reviewer Concerns:**

While a rebuttal might have clarified notation and the "biased gradient" concept, the most critical concerns remain outstanding. Specifically, the exclusive reliance on a single synthetic dataset, the omission of key baselines (e.g., FedSage, FED-PUB), and the theoretical disconnect regarding activation functions were not, and likely could not be, adequately resolved.

**Reviewer Scores:**

Reviewer xVhn (Score: 4): Likely remains unchanged; the structural issue of relying solely on synthetic data and the simplified privacy analysis (ignoring end-to-end accounting) would persist.

Reviewer LLUi (Score: 6): Likely lowers score to 4; exposure to the other reviewers' points regarding missing baselines and the theoretical gap (smoothness vs. ReLU) would likely dampen the initial positive assessment.

Reviewer VT7c (Score: 2): Remains unchanged; the fundamental conflict between the theoretical requirement for smooth activations and practical GNN usage is a hard blocker.

Reviewer e5Xn (Score: 2): Remains unchanged; the absence of comparisons to relevant literature and the weak experimental breadth are sufficient grounds for maintaining a reject.

---

### Decision · Program_Chairs · 2026-01-26

Reject